# Protective Effects of Phycobiliproteins from *Arthrospira maxima* (Spirulina) Against Cyclophosphamide-Induced Embryotoxicity and Genotoxicity in Pregnant CD1 Mice

**DOI:** 10.3390/ph18010101

**Published:** 2025-01-15

**Authors:** Yuliana García-Martínez, Amparo Celene Razo-Estrada, Ricardo Pérez-Pastén-Borja, Candelaria Galván-Colorado, Germán Chamorro-Cevallos, José Jorge Chanona-Pérez, Oscar Alberto López-Canales, Hariz Islas-Flores, Salud Pérez-Gutiérrez, Joaquín Cordero-Martínez, José Melesio Cristóbal-Luna

**Affiliations:** 1Departamento de Farmacia, Escuela Nacional de Ciencias Biológicas, Instituto Politécnico Nacional, Av. Wilfrido Massieu 399, Mexico City C.P. 07738, Mexico; ygarciamart@hotmail.com (Y.G.-M.); arazoe@ipn.mx (A.C.R.-E.); rpastenb@ipn.mx (R.P.-P.-B.); cgalvanc2001@alumno.ipn.mx (C.G.-C.); gchamcev@yahoo.com.mx (G.C.-C.); 2Laboratorio de Micro y Nanobiotecnología, Departamento de Ingeniería Bioquímica, Escuela Nacional de Ciencias Biológicas, Instituto Politécnico Nacional, Av. Wilfrido Massieu 399, Mexico City C.P. 07738, Mexico; jchanona@ipn.mx; 3Departamento de Fisiología, Facultad de Medicina, Universidad Nacional Autónoma de México, Mexico City C.P. 04510, Mexico; oscaralberto_123@hotmail.com; 4Laboratorio de Toxicología Ambiental, Facultad de Química, Universidad Autónoma del Estado de México, Paseo Colón Intersección Paseo Tollocan, Colonia Residencial Colón, Toluca C.P. 50120, Mexico; hislasf@uaemex.mx; 5Departamento de Sistemas Biológicos, Universidad Autónoma Metropolitana-Xochimilco, Calzada del Hueso 1100, Del. Coyoacán, Mexico City C.P. 04960, Mexico; msperez@correo.xoc.uam.mx; 6Laboratorio de Bioquímica Farmacológica, Departamento de Bioquímica, Escuela Nacional de Ciencias Biológicas, Instituto Politécnico Nacional, Mexico City C.P. 11340, Mexico

**Keywords:** phycobiliproteins, cyclophosphamide, teratogenesis, genotoxicity, oxidative stress, antioxidant therapy, pregnant mice, chemotherapy-induced damage, complementary cancer therapy

## Abstract

**Background/Objectives**: In recent years the global incidence of cancer during pregnancy is rising, occurring in 1 out of every 1000 pregnancies. In this regard, the most used chemotherapy drugs to treat cancer are alkylating agents such as cyclophosphamide (Cp). Despite its great efficacy, has been associated with the production of oxidative stress and DNA damage, leading to embryotoxicity, genotoxicity, and teratogenicity in the developing *conceptus*. Therefore, this study aimed to investigate the protective role of phycobiliproteins (PBP) derived from *Arthrospira maxima* (spirulina) in reducing Cp-induced embryotoxicity and genotoxicity in pregnant CD1 mice. **Methods**: Pregnant CD1 mice were divided into five groups: control, Cp 20 mg/kg, and three doses of PBP (50, 100, and 200 mg/kg) + Cp co-treatment. PBP were administered orally from day 6 to 10.5 *dpc*, followed by a single intraperitoneal dose of Cp on 10.5 *dpc*. Embryos were collected at 12.5 *dpc* to assess morphological development and vascular alterations, while maternal DNA damage was evaluated using micronucleus assays and antioxidant enzyme activity in maternal plasma. **Results**: PBP exhibited a dose-dependent protective effect against Cp-induced damage. The 200 mg/kg PBP dose significantly reduced developmental abnormalities, micronucleated polychromatic erythrocytes, and oxidative stress, (as evidenced by increased SOD and GPx activity). **Conclusions**: Phycobiliproteins from *Arthrospira maxima* (spirulina) effectively reduced Cp-induced morphological and vascular alterations in embryos and genotoxicity in pregnant mice. These findings highlight their potential as a complementary therapy to mitigate teratogenic risks during chemotherapy. Further research is needed to optimize dosing and explore clinical applications.

## 1. Introduction

Pregnancy is a multifaceted biological and physiological state that supports the development of a fertilized egg into a fetus within a woman’s uterus, beginning at conception and lasting around 266 days until childbirth [1]. Unlike polyparous species such as mice, who can have multiple offspring in a single litter and have a gestation period of only 19 to 21 days, humans usually experience singleton pregnancies. Not to mention that mice do not experience spontaneous abortion; instead, they reabsorb non-viable embryos [2]. Understanding these and other unique aspects of gestation in mice is the basis for the study of alterations like embryotoxicity and genotoxicity in humans through murine models. During pregnancy, the mother’s body undergoes significant physiological changes that affect all organ systems and the developing fetus. So, it is crucial for healthcare providers to recognize these physiological changes to offer optimal care for the mother and the fetus [3]. Common diseases that may arise during pregnancy include infections, heart disease, gestational diabetes, and cancer, the latter being particularly concerning due to its prevalence and complexity of treatment during pregnancy [4].

Cancer is a disease characterized by the uncontrolled proliferation of transformed cells [5]. One of its most recent definitions states that “cancer is a disease of uncontrolled proliferation by transformed cells subject to evolution by natural selection” [6]. According to data from the International Agency for Research on Cancer (IARC), in 2022 nearly 20 million new cancer cases were reported worldwide, with approximately 9.7 million cancer-related deaths. The most diagnosed cancers were lung cancer (12.4%), breast cancer (11.6%), colorectal cancer (9.6%), prostate cancer (7.3%), and stomach cancer (4.9%) [7]. Although previously rare, the incidence of cancer in pregnant women has increased due to improved diagnostic methods and rising maternal age at conception. It is estimated that cancer affects 1 in 1000 pregnancies [8]. In Mexico, it is estimated that around 2000 to 3000 cases of cancer are diagnosed in pregnant women each year [9]. The most common malignancies diagnosed during pregnancy include breast cancer, cervical cancer, ovarian cancer, melanoma, lymphoma, and thyroid cancer. Similarly, systemic lupus erythematosus [10], Crohn’s disease [11], and sclerosis [12] affects women predominantly in their reproductive years, posing significant risks to both maternal and fetal health.

Pregnancy associated with cancers and autoimmune disease can lead to adverse birth outcomes such as preterm birth, low birth weight, growth restrictions, and a wide range of birth defects affecting the central nervous system [13]. This highlights the necessity of aggressive anticancer treatments like nitrosoureas, antimetabolites, mitosis inhibitors, and alkylating agents like cyclophosphamide (Cp) [14]. As a prodrug, Cp is particularly effective in treating various types of cancers by interfering with DNA replication and cell division [15]. Additionally, Cp is employed to suppress the immune system, making it a versatile and essential medication in modern medical practice [16].

Once in the body, Cp is activated in the liver via cytochrome P450-mediated 4-hydroxylation, producing 4-hydroxycyclophosphamide (4-OHCP) [17]. 4-OHCP rapidly equilibrates with aldophosphamide, which undergoes β-elimination to form the phosphoramide mustard (PM), the DNA-crosslinking therapeutic agent [18] (Figure 1). This bioactivation process also generates acrolein (Acr), a toxic aldehyde associated with bladder damage and hemorrhagic cystitis [15].

Despite its beneficial biological properties, Cp causes adverse effects in pregnant women and fetuses; nausea, hair loss, bone marrow suppression, hepatotoxicity [19], cardiotoxicity, neurotoxicity, immunosuppression [20], and birth defects [13]. The latter is of particular interest to this study, although Cp-induced teratogenesis involves complex mechanisms such as caspases activation or NF-kappaB suppression, the main mechanisms responsible are associated with DNA damage by bioactivation of toxic metabolites and induction of oxidative stress. PM forms covalent bonds with DNA, causing crosslinking, replication errors, and double-strand breaks, while Acr reacts with cellular components, forming Acr-deoxyguanosine adducts that inhibit DNA repair and induce damage through replication interference, mutagenesis, and apoptosis [21,22,23]. If the damage is extensive and cannot be repaired, it can trigger apoptosis or propagation of genetic defects, disrupting embryonic development and causing malformations [23].

On the other hand, Cp induces oxidative stress through reactive oxygen species (ROS) generated during its bioactivation and via angiotensin II overproduction [24]. At same time, Acr exacerbates this by depleting cellular antioxidants like glutathione, leading to mitochondrial dysfunction and increased DNA damage [25]. This oxidative stress activates pro-inflammatory pathways, disrupting embryonic growth and differentiation. Combined with DNA damage, it results in teratogenic effects, including growth restriction, craniofacial and skeletal malformations, and soft tissue defects like hydrocephalus and aphakia [13,26,27].

Projections based on demographic trends suggest that the number of new cancer cases could reach 35 million by 2050 [7]. As a result, the risk that many pregnant women or women of reproductive age are at is palpable. Therefore, it is essential to investigate alternatives that can help prevent or reduce the unwanted effects of Cp. In this context, *Arthrospira maxima*, spirulina (Am), a photosynthetic cyanobacterium that is widely consumed as a nutraceutical is particularly notable for its substantial protein content, 50–70% of its dry weight [28]. This blue-green alga is also rich in micronutrients (thiamine, riboflavin, niacin, cobalamin, folic acid, and A, E, and K vitamins), which are considered valuable tools to counteract the action of free radicals in cells [29]. Additionally, Am contains a variety of essential minerals and bioactive compounds, essential fatty acids and antioxidants like flavonoids, terpenes, phenolic compounds, and pigments such as carotenoids, chlorophylls, and phycobiliproteins (PBP) [30]. Research has shown that Am offers numerous health benefits, including antioxidant, immunostimulatory, antineoplastic, and antiviral properties, as well as a significant detoxifying effect against pollutants like chemicals and heavy metals [31]. Although Am contains a wide variety of bioactive compounds and pigments, multiple studies have demonstrated that one of its most crucial components responsible for its biological effects are the PBP. These protein complexes, in addition to serving as photopigments, have shown significant protective effects against oxidative damage and a low acute toxicity in mice (LD_50_ > 2000 mg/kg) [32].

In this sense, the investigation of antioxidant agents is promising in the context of cancer treatment during pregnancy, especially due to the adverse effects of alkylating drugs such as Cp, which, although effective, generate ROS which subsequently causes oxidative stress. Since this oxidative damage is closely related to embryotoxicity and DNA damage, it is necessary to explore alternative antioxidants capable of mitigating the oxidative stress and toxic effects derived from the use of Cp. Therefore, the ability of PBP to reduce oxidative stress suggests that its inclusion in treatment regimens could offer a complementary approach to protect both mothers and their developing fetuses from damage derived from Cp therapy. Therefore, this study aimed to investigate the protective effect of PBP, protein pigments with broad antioxidant capacity, isolated from Am against Cp-induced damage during the embryonic development of CD1 mice, as well as their antigenotoxic effect in the mothers.

## 2. Results

### 2.1. Phycobiliproteins Extraction and Characterization

The extraction of PBP from Am using thermal shock cryofracturing, centrifugation, and lyophilization had a yield of 23.41%. In addition, the major component extracted was C-PC, APC and in minor amount PE. With a relative purity of 0.88 ± 0.03, 0.39 ± 0.01, and 0.40 ± 0.01, respectively (Table 1).

### 2.2. Results of Teratogenic Test

The administration of the treatments did not result in significant alterations in the weight gain of pregnant females between gestational days 6.5 and 10.5, with all treated groups showing consistent weight increases (Figure 2). During this period, no post-treatment signs of toxicity were observed in females that received PBP as regards its physical appearance and behavior. However, it is important to note that following the administration of Cp at 10.5 *dpc* lethargy, reduced motor activity, and occasional diarrhea were observed in pregnant females. On gestational day 12.5, a weight reduction of nearly 3 g was observed in the group treated solely with Cp at 20 mg/kg (30.94 ± 0.23 g) compared to the control group (33.87 ± 0.28 g). Conversely, the groups pretreated with PBP exhibited a dose-dependent attenuation of weight loss, with the 200 mg/kg dose of PBP maintaining weight gain at levels comparable to the control, and the 50 mg/kg dose resulting in a weight similar to the group Cp 20 mg/kg (34.47 ± 0.26 g and 31.08 ± 0.31 g, respectively).

In addition to the remarkable irregularities in the weight gain of pregnant females, the administration of Cp resulted in notable damage in embryonic development. At 12.5 *dpc*, the embryos of the control group showed a normal development characterized by a prominent cephalocaudal curvature, in which the head is bent towards the trunk. The semi-transparent skin allows the visualization of some internal organs such as the liver and heart, which are well developed. As well as the underlying blood vessels, which are well defined and developed at the cephalic level. And in parallel, we can see that the brain has been divided into three main vesicles (forebrain, midbrain, and hindbrain). Externally, the well-defined optic and otic vesicles are clearly visible; the formation of the face has begun showing outlines of the jaws and well-defined branchial arches. In the trunk, the forelimbs and hindlimbs are not rounded but have developed digital outlines (fingers). And the limbs show a clear segmentation in arms, forearms, thighs, and legs (long bones are present), without vascular alterations or hemorrhages (Figure 3a).

In contrast, embryos treated only with Cp 20 mg/kg show poor embryonic development evidenced by a significantly smaller and less developed body than the control embryos. They were also severe alterations in the cephalic vascular network showing dilatation, disorganization, poor development, and possible punctate hemorrhages. In parallel, although the three main cerebral vesicles are observed, they are poorly developed and differentiated. On the other hand, optic and otic vesicles, although well defined, present vascular alterations and hemorrhages in their periphery. Something similar is observed in the embryonic trunk and in the upper and lower limbs; where in addition to less structural development, the vascular patterns are observed disordered and with hemorrhages in the somites or in the periphery of the plates of the upper and lower limbs. Finally, mandibular outlines and facial development, in general, are severely altered (Figure 3b).

For their part, embryos exposed to PBP show a clear dose-dependent protective effect against Cp damage. At doses of 50 mg/kg of PBP, the embryonic alterations are practically the same as in the group that only received Cp (Figure 3c), showing no protection against Cp. While at doses of 100 and 200 mg/kg of PBP, an improvement in the parameters described above is observed (Figure 3d,e). At the highest dose of PBP, the cephalocaudal length and the development of facial structures and upper and lower limbs were practically the same (or with similar development) as those observed in the control group. The most remarkable feature of this analysis is the clear improvement in the vascular network in embryos exposed to 200 mg/kg of PBP, which is very similar to that of the control group, with slight dilatations and microbleeds that can still be seen at the cephalic level or in the periphery of the plates of the upper and lower limbs. The same as compared to Cp 20 mg/kg group are minimal.

Table 2 displays the effects of PBP at different doses on various parameters of embryonic development that include vascular network of the yolk sac, cephalo-caudal length, number of somites, and several morphological and functional markers. It is observed that Cp treatment caused significant alterations in embryonic development compared to the control group. The vascular network of the yolk sac, embryo rotation, caudal extension, and cephalo-caudal length were all notably reduced in the Cp-treated group, resulting in a decrease in cephalo-caudal length decrease from 9.11 mm (control) to 6.27 mm (Cp-treated). The number of somites also dropped drastically from 51.13 in the control group to 37.63, indicating a developmental delay. These negative effects were also observed in the primitive brain, heart development, and pharyngeal arches, showing significant disruptions. Conversely, co-treatment with PBP showed a protective effect, mitigating the teratogenic effects of Cp in a dose-dependent manner. The highest dose of PBP (200 mg/kg) demonstrated near-complete recovery in most developmental parameters evaluated. For example, the cephalo-caudal length in the PBP 200 mg/kg + Cp group (9.18 mm) almost matched that of the control group (9.11 mm). Similarly, the number of somites in the PBP 200 mg/kg group (51.81) was comparable to the control (51.13), while lower doses (50 and 100 mg/kg) showed a gradual improvement in somite numbers (38.93 and 42.12, respectively), though still below control values. The heart rate measured in beats per minute also reflected the protective effects of PBP, where the Cp group showed a decrease to 137.19 beats/min compared to the control (147.38 beats/min). Treatment with PBP improved this parameter, with the 100 mg/kg group reaching 148.45 beats/min. In summary, PBP co-treatment at 200 mg/kg, effectively mitigated the adverse effects induced by Cp on embryonic development. The data suggest a dose-dependent protective effect, with higher doses of PBP providing the most significant recovery in embryo morphology and function, approaching normal developmental conditions observed in the control group.

### 2.3. Micronucleus Test

#### 2.3.1. Cytotoxicity and Genotoxic Effects

In the micronucleus test, the polychromatic erythrocytes (PCE) and micronucleated PCE (MN-PCE) frequency in the peripheral blood of pregnant mice exposed to Cp was determined as a measure of cytotoxicity and genotoxicity. Our determinations revealed significant cytotoxic effects induced by Cp on bone marrow cells. As shown in Figure 4, a significant decrease in the number of PCE was observed in the Cp 20 mg/kg group, starting from a baseline level at 0 h of exposure (63 ± 2.93) and progressively decreasing at 24 h (39.2 ± 2.87) until reaching its lowest number at 48 h (26.1 ± 2.06) following its administration. Compared to the control group, which maintained a constant number of PCE at the three sampling times (61.1 ± 3.05, 60.3 ± 3.08, and 60.8 ± 3.04, respectively), these results indicate that Cp significantly impaired the proliferation and/or survival of bone marrow cells, a hallmark of cytotoxicity. Interestingly, co-treatment with PBP demonstrated a protective effect against Cp-induced cytotoxicity. The 200 mg/kg dose maintained PCE numbers with minor reductions at all sampling times (62.6 ± 2.79, 56.1 ± 2.99, and 46 ± 2.55), compared to the control group. In contrast, the 50 mg/kg dose showed no protective effect, with values at the three sampling times similar to those of the group that received only Cp (63.5 ± 3.00, 39.5 ± 2.66, 26.2 ± 1.96).

#### 2.3.2. Genotoxic Effect

Parallel to cytotoxicity the micronucleus assay revealed significant genotoxic effects induced by Cp on bone marrow cells. As shown in Figure 5, treatment with Cp resulted in a significant increase in micronucleus frequency compared to the control group, starting from a baseline level at 0 h of exposure (3.5 ± 0.40), increasing at 24 h (35.8 ± 3.00), and reaching its highest frequency of micronuclei at 48 h post-treatment (43.2 ± 3.18). In contrast, the control group maintained a constant micronucleus frequency at all sampling times (3.7 ± 0.60, 3.4 ± 0.39, 3.2 ± 0.60, respectively). These findings indicate Cp-induced DNA damage and chromosomal instability in bone marrow cells. Co-treatment with PBP significantly reduced the frequency of micronuclei induced by Cp, suggesting a protective effect against Cp-induced genotoxicity. The decrease in micronucleus frequency was dose-dependent, with the highest dose of PBP (200 mg/kg) providing greater protection at 24 and 48 h post-treatment by reducing the increase in MN frequency to 9.1 ± 1.16 and 17.2 ± 1.09, respectively. The time-dependent increase in micronucleus frequency in the Cp-treated groups highlights the progressive nature of the genotoxic damage. The initial genotoxic effects were most pronounced at 24 h, and the persistent increase at 48 h suggests continued DNA damage or impaired repair mechanisms.

Table 3 illustrates the percentage of damage reduction (% DR) in cytotoxicity and genotoxicity induced by Cp and mitigated by pre-treatment with PBP at doses of 50, 100, and 200 mg/kg. The results are shown for two time points: 24 h (T2) and 48 h (T3) post-treatment. PBP co-treatment demonstrated a dose-dependent protective effect against both cytotoxic and genotoxic damage. At 24 h post-treatment, the group receiving 200 mg/kg of PBP exhibited the highest reduction in cytotoxic damage, achieving an 80.09% reduction in PCE per 1000 cells, followed by 100 (50.23%) and 50 mg/kg PBP (1.42%). This trend continued at 48 h, where the 200 mg/kg dose again showed the greatest protective effect (57.34%), compared to 100 mg/kg (40.92%), and 50 mg/kg (0.28%).

For genotoxicity, measured as the reduction in MN-PCE per 1000 PCE, a similar dose-dependent pattern was observed. At 24 h, the highest reduction in genotoxic damage was observed in the 200 mg/kg PBP group (81.40%), followed by 100 (46.03%) and 50 mg/kg (4.26%). At 48 h, the 200 mg/kg dose maintained the most significant reduction (64.67%), followed by 100 (34.82%) and 50 mg/kg (7.21%). In summary, the 200 mg/kg PBP dose consistently provided the greatest protection against both cytotoxic and genotoxic damage at 24 and 48 h, confirming a dose-dependent protective effect.

### 2.4. Results of Antioxidant Enzymatic Activity

Results shown in Table 4 display the activity of SOD and GPx in the plasma of pregnant CD1 mice subjected to different treatments. The control group exhibited high SOD activity (4.16 ± 0.041 U/mg protein) and correspondingly high GPx activity (416.33 ± 24.11 U/mL). Cyclophosphamide treatment (Cp 20) significantly reduced the activity of both enzymes, with SOD dropping to 1.88 ± 0.044 U/mg protein and GPx to 197.13 ± 28.24 U/mL, indicating oxidative stress induced by the drug. The pretreatment with PBP at different doses mitigated the oxidative damage caused by Cp. At the lowest dose of PBP (50 mg/kg), there was a slightly increased SOD and GPx activity. A more pronounced recovery was observed with 100 mg/kg PBP, where SOD rose to 3.13 ± 0.055 U/mg protein and GPx to 348.55 ± 19.99 U/mL. The highest dose of PBP (200 mg/kg) restored the enzyme activities to levels comparable to or slightly higher than the control group, with SOD at 4.30 ± 0.031 U/mg protein and GPx at 434.81 ± 23.41 U/mL. These results suggest that phycobiliproteins offer a dose-dependent protective effect against cyclophosphamide-induced oxidative damage in pregnant mice, as evidenced by the restoration of SOD and GPx activities.

## 3. Discussion

This year a study reported that 1 in every 1000 pregnant women worldwide has been diagnosed with some form of cancer, and the number appears to be increasing [8]. Such situations present pregnant women diagnosed with cancer with a challenging dilemma: to undergo treatments to improve their health such as chemotherapy, which can harm the developing fetus, or to prioritize the health of the baby by delaying their treatments and putting their health at risk. In addition, considering that chemotherapy options have become more diverse and complex in recent years, the decision-making process in these cases has become more challenging. This fact has brought increased attention to the concept of shared decision-making (SDM), which is gaining popularity within the medical community as a collaborative approach to addressing these difficult choices [33]. Thus, the present study investigated PBP as an alternative or complementary therapy to help decrease Cp-induced embryotoxicity and genotoxicity during gestation. Our results demonstrated that PBP co-treatment mitigated the negative effects of Cp in a dose-dependent manner, with the highest dose (200 mg/kg) showing the greatest protective effects. These findings could serve as a foundation for developing new complementary cancer therapies.

In our study, pregnant mice exposed to PBP did not show signs of toxicity until 10.5 *dpc* when Cp administration produced lethargy. At 12.5 *dpc*, weight loss was observed in the Cp-exposed groups, mainly in those who received the lowest dose of PBP. Although it was observed that the administration of Cp caused lethargy in pregnant females and some episodes of diarrhea, as well as discomfort that could have been reduced the food consumption, our results indicate that the decrease in weight gain was due to a delay in embryonic development observed in the *conceptus*. The consequences of prenatal exposure to Cp-induced embryopathy are characterized by intrauterine growth restriction, developmental delays, a wide range of congenital malformations, and intrauterine death through oxidative stress and DNA damage [13,34]. Notably, PBP co-treatment effectively preserved maternal weight gain and prevented embryonic malformations, reduced cephalo-caudal length, decreased the number of somites and neural tube defects, among others, which results in a normal embryonic development similar to that of the control group. These findings are supported by research that highlights the antioxidant properties of PBP in reducing oxidative stress and preventing damage to developing tissues [35,36].

In accordance with the literature, the protective effects of PBP against Cp-induced embryotoxicity are attributed to four main mechanisms: neutralization of free radicals, reduction in oxidative stress, cell protection, and to a lesser extent, modulation of immune response. The strong ability of PBP to inhibit free radicals is of great interest because they can neutralize and sequester a wide variety of ROS and reactive nitrogen species (NRS) due to its chemical structures like phycobilins (linear tetrapyrrole chromophores) and α and β subunits [37]. Such neutralization is complemented by the chelating properties of PBP, which allow them to trap ferrous ions and reduce oxidative stress more efficiently [38]. Some authors have shown that another way in which PBP can reduce oxidative stress is decreasing the production of ROS [39], and at the same time increasing the activity of antioxidant enzymes such as SOD, CAT, or GPx, that play a vital role in neutralizing the ROS generated [36]. In the same way, it has been reported that PBP protects against cell damage thanks to its ability to delay or inhibit protein oxidation, lipid peroxidation, and DNA degradation [40]. In addition, some reports have shown that PBP can modulate the immune response, acting on pathways such as NF-κB, TLR, PI3K/Akt/mTOR and Nrf2 to inhibit inflammation [41].

In terms of the protective effects on embryonic development, the improvement observed in PBP-treated groups aligns with the well-established role of antioxidants in protecting against teratogenicity [42]. PBP administration promoted normal development by neutralizing ROS and RNS and enhancing the activity of internal antioxidant defenses, which together helped to maintain a balance between oxidative stress and the organism’s antioxidant capacity, thus protecting both maternal tissues and developing embryos from oxidative damage. Not to mention that PBP can modulate inflammatory processes that could impair embryonic development, and at the same time protects against DNA damage. The latter aspect is of particular importance given the fragility of embryonic processes considering the high rate of DNA replication and cell division required during embryogenesis—DNA replication in embryonic stages occurs 20 to 60 times faster than in somatic cells [43]. It is important to note that the protective response of PBP, in which higher doses show greater protection against Cp-induced damage, is consistent with the literature on antioxidants as they often show greater efficacy at higher concentrations [44].

In addition to their antioxidant, anti-inflammatory, and immunomodulatory properties, PBP offers a unique advantage in embryonic development that not all antioxidant compounds possess: their ability to restore vascular contractility [45]. This effect is of special interest for our research, since functional assays using mesenteric arteries from Cp-treated mice have demonstrated a significant decrease in vascular contractility associated with vascular lipoperoxidation, indicating that Cp causes significant vascular damage [46,47]. As observed when analyzing the main structures of the embryos exposed to Cp, the blood system was one of the most affected. The CNS, somites, eyes, and upper and lower limbs showed severe damage in their blood vessels such as dilatation, microbleeds, and alterations in the pattern of vascular development. Our results indicate that Cp significantly alters embryonic vascular development, thereby impairing normal embryonic growth and maternal health. Consequently, we can infer that the improvement in embryonic development observed in the PBP-treated group is not solely attributable to the reduction in ROS and protection against genetic damage. In contrast, PBP have the ability to protect not only against vascular lipoperoxidation, but can also restore vascular contractility, which is a crucial factor contributing to their antiteratogenic effect against Cp at the CNS level. This aspect warrants further investigation to fully understand its implications and action mechanisms.

In a complementary manner, the results of the micronucleus assay support the protective effect of PBP against Cp, as the frequency of MN-PCE decreased significantly in the PBP-treated groups. Such findings suggest that PBP mitigates the DNA damage caused by Cp, which is known to induce double-strand breaks and chromosomal instability via its active metabolites, PM and Acr [21]. The protective effect of PBP on MN-PCE frequency is likely linked to its ability to reduce oxidative stress, which plays a key role in DNA damage [22]. In the same way, the observed reduction in cytotoxicity, as indicated by the preservation of PCE frequency, further supports the role of PBP in maintaining cellular integrity against Cp-induced damage, possibly through antioxidant mechanisms or by modulating cellular signaling pathways involved in cell death. The observed decrease in the number of PCE as a function of time highlights the dynamic nature of the cytotoxic response. While cytotoxic effects were visible from the first 24 h, the persistent decline at 48 h suggests ongoing cell damage or poor recovery. Overall, these results provide convincing evidence that CP induces cytotoxicity in bone marrow cells, as evidenced by the significant reduction in the number of PCE [48]. And at the same time, co-treatment with PBP appears to alleviate these cytotoxic effects, indicating a protective role of PBP against Cp-induced cytotoxicity.

The dose-dependent antigenotoxic and anticytotoxic capacity in the presence of Cp was previously reported by Oliveira et al. [49], whose research demonstrates that antioxidant compounds such as β-glucan can reduce Cp-induced bone marrow suppression by protecting hematopoietic cells from oxidative damage (significantly reducing MN-PCE frequency and preventing PCE loss) in Cp-treated pregnant females. Our data are consistent with previous research that highlights the ability of antioxidants such as melatonin, β-caryophyllene, carvedilol, L-carnitine, quercetin, etc., to reduce Cp-induced malformations and DNA damage by neutralizing ROS and enhancing cellular repair mechanisms [25,26,50,51,52].

The results of this study suggest that PBP could be a promising candidate for adjuvant therapy in pregnant women undergoing treatment with Cp, either for cancer, autoimmune diseases, nephrotic syndrome, or any other disease for which the treatment includes alkylating agents such as Cp. The ability of PBP to reduce embryotoxicity and genotoxicity resulting from Cp treatment is particularly noteworthy, as it opens the possibility of reducing the teratogenic risks associated with alkylating agent chemotherapy. It also allows the patient to continue with her treatment without endangering the integrity of her developing baby. However, further research is needed to demonstrate that PBP, in addition to reducing the toxic effects of Cp, does not compromise the efficacy of cancer treatment.

Despite these promising findings, it is important to acknowledge certain limitations: the mouse model does not fully replicate human physiological responses to PBP and Cp; although we explored different doses of PBP the optimal dosing regimen for maximizing protective effects while minimizing potential toxicity remains to be fully established; PBP were shown to reduce the toxic effects of Cp, but we do not know whether PBP interfere with the therapeutic capacity of this alkylating agent. Therefore, future studies should also investigate the long-term developmental outcomes in offspring exposed to PBP and Cp during gestation, as well as the optimal dosing regimen for balancing protective effects with potential toxicity. Molecular studies focusing on the specific pathways involved in PBP-mediated protection could provide valuable insights into its therapeutic potential. Among these, it would be interesting to explore its ability to restore vascular contractility, and to determine whether PBP interferes with the therapeutic effects of Cp.

The present study reinforces the notion that PBP, potent natural antioxidant, has a great potential to be utilized as a complementary strategy alongside existing chemotherapy regimens. By mitigating oxidative stress and enhancing cellular protection, PBP could improve treatment outcomes and reduce the adverse effects associated with conventional chemotherapeutic agents.

## 4. Materials and Methods

### 4.1. Aqueous Extract of Arthrospira Maxima (Spirulina)

A total of 5 g of Am powder (generously donated by Alimentos Esenciales para la Humanidad S.A. de C.V., Ciudad de México, Mexico) were suspended in 20 mL of phosphate-buffered saline (PBS) (20 mmol, pH 7.4). The suspension underwent two cycles of freezing (−70 °C for 2 h) and thawing (37 °C for 30 min), followed by centrifugation at 13,000 rpm × 30 min using a JA-17 rotor in a Beckman Coulter centrifuge (Beckman Coulter, Inc., California, CA, USA) for two cycles to ensure the purity of the PBP by separating the characteristic dark blue, PBP-rich supernatant from the cell precipitate. The aqueous extract was analyzed spectrophotometrically to assess its purity and subsequently lyophilized (FreeZone 4.5, Labconco, Kansas City, MO, USA) at −50 °C and 0.0133 mBar for 36 h until a fine light blue powder was obtained under dark conditions. The lyophilized extract was then stored at −70 °C, protected from light until its use [53].

The optical densities (O.D.) of the aqueous extract rich in PBP were measured at specific wavelengths (615, 652, and 562 nm) using a Shimadzu BioSpec-mini DNA/RNA/Protein UV-Visible Spectrophotometer (Shimadzu Corporation, Kyoto, Japan), in order to determine the absorbance of C-phycocyanin (C-PC, λ_max_ = 610–620 nm), allophycocyanin (APC, λ_max_ = 650–655 nm), and phycoerythrin (PE, λ_max_ = 540–570 nm), respectively [54]. The concentration of three PBP were calculated following the method described by Bennett and Bogorad [55], with the following equations:$$\text{C-PC}\left(\frac{\mathrm{mg}}{\mathrm{mL}}\right)=\frac{A_{615}-0.474\left(A_{652}\right)}{5.34}$$$$\mathrm{APC}\left(\frac{\mathrm{mg}}{\mathrm{mL}}\right)=\frac{A_{652}-0.208(A_{615})}{5.09}$$$$\mathrm{PE}\left(\frac{\mathrm{mg}}{\mathrm{mL}}\right)=\frac{A_{562}-2.41\left[\text{C-PC}\right]-0.849[\mathrm{APC}]}{9.62}$$

The relative purity was estimated using the method described by Boussiba and Richmond [53,54,56] with the following equations:$$\text{C-PC}\;\mathrm{purity}=\left(\frac{A_{615}}{A_{280}}\right),\;\;\;\;\;\;\mathrm{APC}=\left(\frac{A_{652}}{A_{280}}\right),\;\;\;\;\;\;\mathrm{PE}=\left(\frac{A_{562}}{A_{280}}\right)$$

The extraction of the PBP was estimated following the equation proposed by Silveira et al. [57], where C-PC, APC, and PE are expressed as mg/mL, V is the solvent volume (mL), and DB is the dry biomass (g):$$\mathrm{PBP}\;\mathrm{Yield}\;\left(\frac{\mathrm{mg}}{g}\right)=\frac{\left(\text{C-PC}+\mathrm{ACP}+\mathrm{PE}\right)\left(V\right)}{\mathrm{DB}}$$

### 4.2. Teratogenic Test

#### 4.2.1. Animals and Housing Conditions

A total of 10 sexually mature male and 75 nulliparous female CD1 mice aged 8 weeks were obtained from the breeding colony of the Laboratory Animal Production and Experimentation Unit (UPEAL) of the Universidad Autónoma Metropolitana (UAM) Xochimilco, located in Mexico City, Mexico. All animals were housed in polycarbonate cages with sterile pinewood shavings as bedding in an air-conditioned room under controlled conditions; 23 ± 2 °C, 55–60% relative humidity, and a 12 h light/dark cycle (lights on at 10:00 h). They received a standard diet (Rodent Lab Chow 5001, Purina, St. Luis, MO, USA) and access to purified water *ad libitum*, and were acclimatized for 2 weeks prior to the initiation of the experiment. Approval for the study was obtained from the Research Ethics Committee of the Escuela Nacional de Ciencias Biológicas (CEI-ENCB) under protocol number 09-CEI-002-20190327. All procedures and animal handling were conducted in strict adherence to the Mexican Official Standard “NOM-062-ZOO-1999” concerning technical statements for production, care, and use of experimental laboratory animals [58].

#### 4.2.2. Study Design and Treatment Administration

Females were housed three per cage and males housed individually, after 2 weeks of acclimatization animals were coupled during the last 3 h of the dark phase of their light/dark cycle. This was performed with the aim of having better control over the moment of fertilization. At the end of the dark phase, the vaginal cavity of each female was examined, the presence of a vaginal plug was taken as evidence that mating had occurred, and that day was designated as *day post coitum* 0.5 (0.5 *dpc*), also referred to by some researchers as gestation day 0 (GD0).

Each mated female was assigned a unique identification number through coat staining and randomly allocated to one of five groups (*n* = 15): control (vehicle, purified water), Cp 20 mg/kg, and three different doses of PBP: 50, 100, and 200 mg/kg. Similarly, Cp doses were selected based on data in the literature that ensured the induction of malformations in developing embryos of mice [59]. All solutions were freshly prepared before its administration; all treatments, except for Cp, were administered by gavage in a constant volume of 10 mL/kg from 6.5 to 10.5 *dpc*, at the same time once a day. On the other hand, Cp was administered as a single intraperitoneal (IP) dose only on 10.5 *dpc* in the same constant volume. On 12.5 *dpc*, pregnant females were euthanized by cervical dislocation and the uterine horns were removed through a transverse abdominal incision. The position of every embryo on the uterine horns was registered, and the embryos were dissected and analyzed for external morphological alterations, modifications in the embryonic vasculature, and antioxidant enzyme activity.

#### 4.2.3. Embryonic Morphological Analysis

Embryos were carefully extracted and placed in cold PBS, pH = 7.4, to measure crown-rump length and develop a morphological and structural analysis of alterations using a Wild Heerbrugg M7S stereo microscope (Leica Microsystems, Heerbrugg, Switzerland). A descriptive value was assigned based on the dysmorphology scoring system proposed by Zhang et al. [60].

#### 4.2.4. Statistical Analysis

Maternal body weight was analyzed using two-way ANOVA for repeated measures *post hoc* Student–Newman–Keuls test. Somatometric variables were analyzed using one-way ANOVA followed by a *post hoc* Student–Newman–Keuls test. Implantation and embryonic parameters (live or dead embryos, heartbeats, number of somites, development scoring, etc.) were evaluated using ANOVA on ranks (Kruskal–Wallis test), and maldevelopment was evaluated using a Chi-square test followed by Fisher’s exact test. All data were processed using the statistical software SigmaPlot 15.0, with statistical significance set at *p* ≤ 0.05.

### 4.3. Micronuclei Assay

Micronuclei evaluation was performed on peripheral blood reticulocytes from pregnant females of the previously described five groups (in the teratogenic test), from which 10 were randomly selected for each group of 15 females (*n* = 10). Peripheral blood samples were collected from the tail of each mouse (5 µL approximately) prior to Cp administration at 10.5 *dpc* (0 h). After Cp administration, the remaining samples were collected every 24 h for the next 48 h. The last blood sample was collected at 12.5 *dpc*, prior to euthanasia. To prepare each smear, a drop of blood was placed at the end of each clean slide and extended (two smears of each sampling time per mouse). Next, cells were fixed in 96% methanol for 3 min, briefly rinsed under running water, dried, and stained with 5% Giemsa (Sigma-Aldrich, St. Louis, MO, USA) in PBS for 10 min [61]. The stained and coded slides were examined under a Zeiss light microscope (Carl Zeiss Microscopy GmbH, Jena, Germany) using a 100× objective to count the number of young PCE and mature normochromatic erythrocytes (NCE) within 1000 cells per mouse at each sampling time, to determine the PCE/NCE ratio as an indicator of the cytotoxicity of the treatments. Additionally, the frequency of MN-PCE was quantified within 1000 PCE per mouse at each sampling time, as a measure of the genotoxic effects of the treatments. Finally, to assess the chemopreventive capacity of PBP against Cp in the micronuclei test, the percentage of damage reduction (%DR) was calculated with the equation proposed by Manoharan and Beneriee [62,63]:$$\%\mathrm{DR}=\left(\frac{{\mathrm{MN}}_{\mathrm{positive}\;\mathrm{control}}-{\mathrm{MN}}_{\mathrm{treated}\;\mathrm{group}}}{{\mathrm{MN}}_{\mathrm{positive}\;\mathrm{control}}-{\mathrm{MN}}_{\mathrm{negative}\;\mathrm{control}}}\right)\left(100\right).$$

#### Statistical Analysis of Micronuclei Assay

The data corresponding to the PCE and MN-PCE were analyzed using an ANOVA on ranks for repeated measures followed by a *post hoc* Student–Newman–Keuls test. All statistical analyses were performed using the statistical software SigmaPlot 15.0, with significance set at *p* ≤ 0.05.

### 4.4. Antioxidant Enzymatic Activity

At 12.5 *dpc*, peripheral blood samples were collected from pregnant females via retro-orbital puncture using heparinized capillary tubes to prevent coagulation. The blood samples were immediately transferred into EDTA-treated Eppendorf tubes to prevent clotting. The samples were centrifuged at 2000× *g* for 10–15 min at 4 °C to separate plasma from the cellular components. Plasma, rather than serum, is ideal for enzymatic antioxidant assays as it preserves the integrity of enzymes like superoxide dismutase (SOD) and glutathione peroxidase (GPx). The plasma was carefully aliquoted to avoid contamination from cellular components and stored at −70 °C until further analysis. The total protein concentration and the activity of antioxidant enzymes SOD and GPx were determined using validated spectrophotometric assays.

The activities of SOD and GPx were determined in collected plasma using commercial kits (RANSOD SD125 and RANSEL RS504, Randox Laboratories Ltd., Crumlin, Co., Antrim, UK). The assays were performed according to the manufacturer’s instructions, with absorbance measured at 505 nm for SOD and 340 nm for GPx. The SOD assay utilizes xanthine and xanthine oxidase to generate superoxide radicals, which react with 2-(4-iodophenyl)-3-(4-nitrophenol)-5-phenyltetrazolium chloride (INT) to form a red formazan dye. SOD inhibits this reaction by dismutating superoxide radicals. SOD activity was expressed as units per mg of protein [64]. The GPx assay is based on the oxidation of glutathione (GSH) by cumene hydroperoxide, catalyzed by GPx. The oxidized glutathione (GSSG) is then reduced by glutathione reductase (GR) in the presence of NADPH, which is oxidized to NADP+. The decrease in absorbance at 340 nm was monitored, and GPx activity was expressed as units per mg of protein [65]. Total protein concentration was measured using the Randox Total Protein Kit (TP245, Randox Laboratories Ltd., Crumlin, Antrim, UK). The assay is based on the Biuret method, in which proteins form a violet-colored complex with copper ions in an alkaline solution. The determination was performed according to the manufacturer’s protocol, with absorbance measured at 546 nm using a spectrophotometer. Bovine serum albumin (BSA) was used as the calibration of the standard for protein quantification [66].

#### Statistical Analysis

Enzymatic activity data were analyzed with a one-way ANOVA *post hoc* Student–Newman–Keuls. All statistical analyses were performed using the statistical software SigmaPlot 15.0, with significance set at *p* ≤ 0.05.

## 5. Conclusions

Cp administration significantly reduced SOD and GPx activities, indicating increased oxidative stress in pregnant CD1 mice. PBP from Am effectively mitigated the embryotoxic and genotoxic effects of Cp in a dose-dependent manner. The highest dose of PBP tested (200 mg/kg) was THE most effective in reducing developmental abnormalities and DNA damage, likely through the enhancement of antioxidant enzyme activity to counter oxidative stress. While multiple mechanisms are involved, one of the most intriguing mechanisms through which PBP appears to prevent Cp-induced embryonic damage is through their protective effects against vascular injury. This mechanism merits additional exploration in future studies.

These findings suggest that PBP could be considered a potential adjunct therapy for pregnant women undergoing chemotherapy, providing embryofetal protection without compromising maternal cancer treatment. Further research is needed to ensure that PBP does not interfere with the therapeutic effect of Cp and to establish optimal dosing, safety, and efficacy in clinical settings.

## Figures and Tables

**Figure 1 pharmaceuticals-18-00101-f001:**
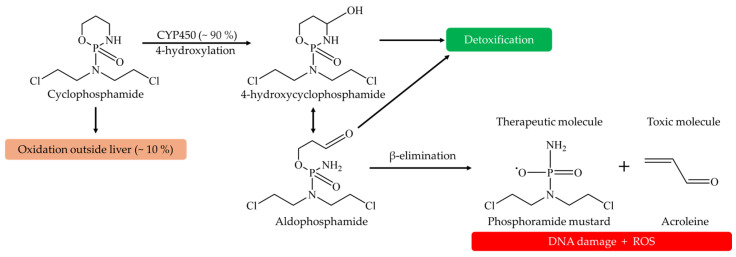
Schematic representation of the hepatic biotransformation of cyclophosphamide. Metabolic activation leads to the formation of the phosphoramide mustard and acrolein.

**Figure 2 pharmaceuticals-18-00101-f002:**
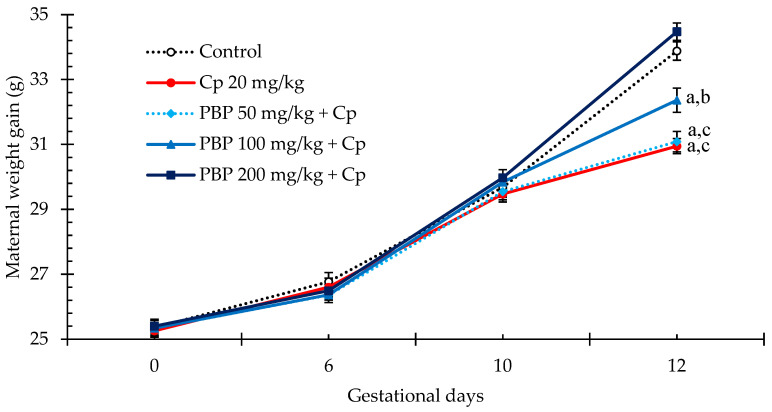
Weight gain of pregnant females from 0.5 to 12.5 *dpc*. Each point represents the mean ± SEM (*n* = 15). PBP, phycobiliproteins; Cp, cyclophosphamide. Two-way ANOVA for repeated measures *post hoc* Student–Newman–Keuls revealed significant differences (*p* ≤ 0.05) vs. ^a^ Control, ^b^ Cp 20 mg/kg, and ^c^ PBP 200 mg/kg on 12.5 *dpc*.

**Figure 3 pharmaceuticals-18-00101-f003:**
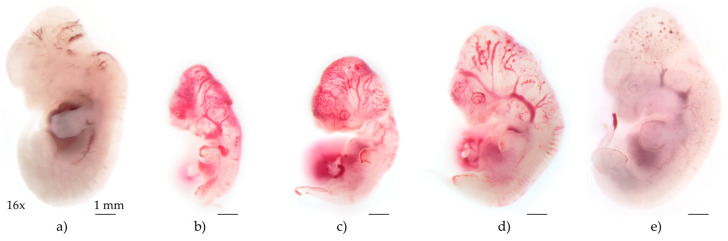
Dose-dependent protective effects of PBP against Cp-induced teratogenicity in CD1 mouse embryos at 12.5 *dpc*. All images were taken on a Wild Heerbrugg M7S stereo microscope at 16x magnification, black bars represent 1 mm. (**a**) Control embryo shows normal development of facial structures, upper and lower limbs, and blood vessels; (**b**) embryos exposed to Cp 20 mg/kg are small, they show delayed embryonic development, disordered somites (lumbar, sacral, and coccygeal), as well as dilatation and disorganization of the blood vessels (mainly at the level of the brain, eyes and limbs); (**c**) embryos treated with PBP 50 mg/kg + Cp show the similar type and severity of developmental and vasculature alterations as those shown by embryos exposed to Cp alone; (**d**) embryos treated with PBP 100 mg/kg + Cp show better embryonic development, the upper and lower limbs have less vascular alterations and better development. Although dilatation of the encephalic blood vessels is still observed, it is more organized (probably due to the more advanced development of the embryo). In parallel, although the somites show a better organization, they present alterations in their vasculature; (**e**) embryos treated with PBP 200 mg/kg + Cp show a practically normal embryonic development, with slight blood vessels alterations at the encephalic level in the form of small dilatations. The somites are perfectly organized and there are no vascular alterations.

**Figure 4 pharmaceuticals-18-00101-f004:**
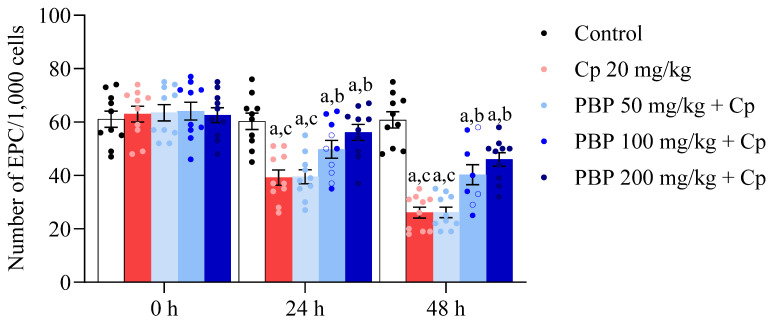
Number of polychromatic erythrocytes found in the peripheral blood of pregnant treated mice. Each bar represents the mean ± SEM of PCE in 1000 cells per mouse (*n* = 10) in 10.5 (0 h), 11.5 (24 h) and 12.5 (48 h) *dpc*. In addition, points in each bar represent the dispersion of the individual data. Polychromatic erythrocytes, PCE; cyclophosphamide, Cp; phycobiliproteins, PBP. ANOVA on ranks for repeated measures *post hoc* Student–Newman–Keuls revealed significant differences (*p* ≤ 0.05) vs. ^a^ Control, ^b^ Cp 20 mg/kg, and ^c^ PBP 200 mg/kg.

**Figure 5 pharmaceuticals-18-00101-f005:**
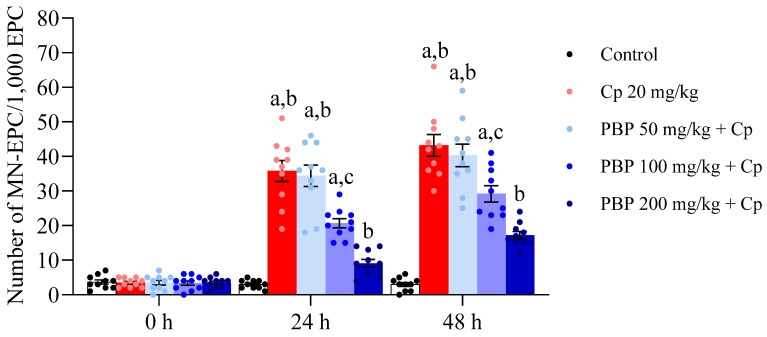
Number of micronucleated polychromatic erythrocytes found in the peripheral blood of pregnant treated mice. Each bar represents the mean ± SEM of MN-PCE in 1000 PCE per mouse (*n* = 10) in 10.5 (0 h), 11.5 (24 h) and 12.5 (48 h) *dpc*. In addition, points in each bar represent the dispersion of the individual data. Micronucleated polychromatic erythrocytes, MN-PCE; polychromatic erythrocytes, PCE; cyclophosphamide, Cp; phycobiliproteins, PBP. ANOVA on ranks for repeated measures *post hoc* Student–Newman–Keuls revealed significant differences (*p* ≤ 0.05) vs. ^a^ Control, ^b^ Cp 20 mg/kg, and ^c^ PBP 200 mg/kg.

**Table 1 pharmaceuticals-18-00101-t001:** PBP extraction, concentration, and relative purity.

PBP	Ratio Biomass (g): Solvent (mL)	Concentration (mg/mL)	Relative Purity
C-PC	1:4	1.77 ± 0.067	0.88 ± 0.03
APC		0.66 ± 0.027	0.39 ± 0.01
PE		0.07 ± 0.001	0.40 ± 0.01

The concentration and relative purity with which the PBP were obtained from the aqueous ex-tract of Am are shown. Data are expressed as mean ± SEM. PBP, phycobiliproteins; C-PC, c-phycocyanin; APC, allophycocyanin; PE, phycoerythrin.

**Table 2 pharmaceuticals-18-00101-t002:** Development score of embryos at 12.5 *dpc*, exposed to PBP and Cp.

	Dosis (mg/kg)				
Embryonic Development	Control	Cp 20	PBP 50 + Cp	PBP 100 + Cp	PBP 200 + Cp
*n*	202	191	208	211	199
Vascular network of the yolk sac	4.91 ± 0.04	4.78 ± 0.07	4.83 ± 0.06	4.88 ± 0.02	4.94 ± 0.05
Caudal extension	4.97 ± 0.03	4.51 ± 0.09 ^a,c^	4.71 ± 0.06 ^a^	4.81 ± 0.11 ^a,b^	4.95 ± 0.03 ^b^
Cephalo-caudal length (mm)	9.11 ± 0.20	6.27 ± 0.19 ^a,c^	6.81 ± 0.18 ^a^	8.01 ± 0.11 ^a,b,c^	9.18 ± 0.09 ^b^
Number of somites	51.13 ± 0.78	37.63 ± 0.93 ^a,c^	38.93 ± 0.98 ^a,c^	42.12 ± 0.74 ^a,b,c^	51.81 ± 0.97 ^b^
Primitive brain	5.00 ± 0.00	3.72 ± 0.12 ^a,c^	3.78 ± 0.13 ^a^	4.66 ± 0.07 ^a,b^	4.91 ± 0.15 ^b^
Heart	5.00 ± 0.00	4.07 ± 0.11 ^a,c^	4.11 ± 0.15 ^a^	4.68 ± 0.08 ^a,b^	4.97 ± 0.05 ^b^
Pharyngeal arches	4.98 ± 0.01	4.21 ± 0.05 ^a,c^	4.11 ± 0.09 ^a^	4.72 ± 0.03 ^a,b^	4.93 ± 0.03 ^b^
Facial morphology	4.98 ± 0.02	3.41 ± 0.14 ^a,c^	3.50 ± 0.11 ^a^	4.56 ± 0.15 ^a,b^	4.77 ± 0.09 ^b^
Beats/min	112.38 ± 8.36	117.19 ± 6.93	123.44 ± 9.89 ^a^	118.45 ± 7.67	112.95 ± 8.41
Maldevelopment (%)	6.4	78.59 ^a,c^	75.34 ^a^	51.18 ^a,b,c^	9.8 ^b^

Data on the development score of embryos exposed to different treatments, 15 litters per group, are shown. The values are based on a dysmorphology scoring system (0–5). Data are expressed as mean ± SEM, and percentage. PBP, phycobiliproteins; Cp, cyclophosphamide; *dpc*, *day post coitum*. ANOVA on ranks (Kruskal–Wallis test) and Chi-square test followed by Fisher’s exact test (for maldevelopment), revealed significant differences (*p* ≤ 0.05) vs. ^a^ Control, ^b^ Cp 20 mg/kg, and ^c^ PBP 200 mg/kg on 12.5 *dpc*.

**Table 3 pharmaceuticals-18-00101-t003:** Reduction in cytotoxic and genotoxic damage in the micronucleus test.

	PCE/1000 Cells (% DR)			MN-PCE/1000 PCE (% DR)		
Treatments	T0 (0 h)	T2 (24 h)	T3 (48 h)	T0 (0 h)	T2 (24 h)	T3 (48 h)
Control	-	-	-	-	-	-
Cp 20 mg/kg	-	-	-	-	-	-
PBP 50 mg/kg + Cp	0	1.42	0.28	0	4.26	7.21
PBP 100 mg/kg + Cp	0	50.23	40.92	0	46.03	34.82
PBP 200 mg/kg + Cp	0	80.09	57.34	0	81.40	64.67

%DR, damage reduction; PCE, polychromatic erythrocytes; MN-PCE, micronucleated polychromatic erythrocytes; PBP, phycobiliproteins; Cp, cyclophosphamide.

**Table 4 pharmaceuticals-18-00101-t004:** Antioxidant enzyme activity in plasma of pregnant CD1 mice treated with cyclophosphamide and phycobiliproteins.

Treatments (mg/kg)	SOD (U/mg Protein)	GPx (U/mg/mL)
Control	4.16 ± 0.041	416.33 ± 24.11
Cp 20	1.88 ± 0.044 ^a,c^	197.13 ± 28.24 ^a,c^
PBP 50 + Cp 20	2.03 ± 0.053 ^a,c^	224.09 ± 17.4 ^a,c^
PBP 100 + Cp 20	3.13 ± 0.055 ^a,b^	348.55 ± 19.99 ^a,b^
PBP 200 + Cp 20	4.30 ± 0.031 ^b^	434.81 ± 23.41 ^b^

Cp treatment significantly reduced antioxidant enzyme activities, whereas co-administration of PBP restored SOD and GPx activities in a dose-dependent manner. SOD activity is expressed as units per mg of protein (U/mg protein), and GPx activity is expressed as units per mL (U/mL). Data are presented as mean ± SEM (*n* = 15). All experiments were performed in triplicate. SOD, superoxide dismutase; GPx, glutathione peroxidase; Cp, cyclophosphamide; PBP, phycobiliproteins. One-way ANOVA *post hoc* Student–Newman–Keuls revealed significant differences (*p* ≤ 0.05) vs. ^a^ Control, ^b^ Cp 20 mg/kg, and ^c^ PBP 200 mg/kg on 12.5 *dpc*.

## Data Availability

The data presented in this study are available upon request from the corresponding author due to privacy considerations.

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
