# Peer review of "Protective Effects of Phycobiliproteins from Arthrospira maxima (Spirulina) Against Cyclophosphamide-Induced Embryotoxicity and Genotoxicity in Pregnant CD1 Mice"

_pharmaceuticals, 2025, doi:10.3390/ph18010101_

Round 1

Reviewer 1 Report

Comments and Suggestions for Authors

The article has a scientific merit and provides readers with new data related to the plant-based compounds and their potential use as the chemopreventive. It connects to modern trends in science in order to use safe products, especially when we take into consideration pregnancy. There is not many research regarding plant products and pregnancy and cancers and chemotherapy. In this case new approaches are needed and this article signs in that conception.

Introduction - please use a correct name - Arthrospira maxima - when mentioned for the first time and then Spirulina.

Results and Discussion: - The results are presented in a clear and readable manner, statistics is included, the appropriate controls were used. The results have been discussed with those available in literature, relevant publications were cited. Generally, there are not many results in this subject and no many researchers deal with that in a mouse model.

Materials and methods The methodology used is described in detail and is relevant to the research questions as well as is adequate and justified. In an experimental part the Author did apply modern analytical approaches, used high quality tools to do all research. These tools meet all criteria in current pre-clinical pharmaceutical sciences.

Conclusion: - This part sums up the meaning of the results obtained and their application in the future as an adjunct in pregnant women undergoing chemotherapy. The Authors point out what more may be done in the future including a potential interaction with cytostatics.

Author Response

Comment 1: The article has a scientific merit and provides readers with new data related to the plant-based compounds and their potential use as the chemopreventive. It connects to modern trends in science in order to use safe products, especially when we take into consideration pregnancy. There is not many research regarding plant products and pregnancy and cancers and chemotherapy. In this case new approaches are needed and this article signs in that conception.

Answer 1: Thank you for your positive feedback regarding the scientific merit of our article. We appreciate your recognition of the importance of exploring plant-based compounds as chemopreventive agents in pregnancy. We agree that there is a need for more research in this area, and we are glad that our study contributes to this important discussion.

Comment 2: Introduction - please use a correct name - Arthrospira maxima - when mentioned for the first time and then Spirulina.

Answer 2: Thank you for your comment, we appreciate your attention to detail. In the revised manuscript, we have ensured that the correct name Arthrospira maxima (spirulina) is used upon its first mention, in the title and in the abstract. This will align with current taxonomic standards and enhance the clarity of our work. Thank you for your valuable suggestion.

Comment 3: Results and Discussion: - The results are presented in a clear and readable manner, statistics is included, the appropriate controls were used. The results have been discussed with those available in literature, relevant publications were cited. Generally, there are not many results in this subject and no many researchers deal with that in a mouse model.

Answer 3: Thank you for your positive feedback, we appreciate your acknowledgment of the clarity, statistical analysis, and appropriate controls used in our study. Your insights reinforce the significance of our findings within this field.

Comment 4: Materials and methods The methodology used is described in detail and is relevant to the research questions as well as is adequate and justified. In an experimental part the Author did apply modern analytical approaches, used high quality tools to do all research. These tools meet all criteria in current pre-clinical pharmaceutical sciences.

Answer 4: Thank you for your encouraging feedback, we are pleased to hear that you found our methodology to be thoroughly described and relevant to our research questions. It is gratifying to know that our application of analytical techniques and tools met the criteria for current pre-clinical pharmaceutical sciences. Your positive remarks are invaluable to us and will certainly contribute to enhancing our manuscript.

Comment 5: Conclusion: - This part sums up the meaning of the results obtained and their application in the future as an adjunct in pregnant women undergoing chemotherapy. The Authors point out what more may be done in the future including a potential interaction with cytostatics.

Answer 5: Thanks, we believe that our conclusion effectively summarizes the significance of our findings and their potential applications as adjunct therapies for pregnant women undergoing chemotherapy. We appreciate your acknowledgment of our discussion regarding future research directions, particularly the exploration of possible interactions between plant-based compounds and cytostatic agents. This aspect is crucial, as understanding these interactions could enhance treatment safety and efficacy for this vulnerable population.

Reviewer 2 Report

Comments and Suggestions for Authors

Abstract. Please the journal instruction for editing the manuscript. To my knowledge the abstract should be without headings. 

Introduction. In my opinion, this section is too long. Also, the protective effect of phycobiliproteins from Arthrospira (Spirulina) maxima against cyclophosphamide-induced embryotoxicity and genotoxicity was studied for pregnant mice. Therefore, some general details about pregnancy in mice should be added in introduction, not only general things about pregnancy in humans. 

L414 -"In accordance with the literature..."at the end of the sentence should be added the reference where the four mechanisms are mentioned.

L498 - "Future studies should also investigate the long-term developmental outcomes in off- spring exposed to PBP and Cp during gestation, as well as the optimal dosing regimen for balancing protective effects with potential toxicity" In my opinion the limitations of the study should be clearly presented as limitations in the main text. I

minor comments: L360 Table X?

Author Response

Comment 1: Abstract. Please the journal instruction for editing the manuscript. To my knowledge the abstract should be without headings.

Answer 1: Thank you for your comment regarding the format of the abstract. We have reviewed the format of the abstract shown in the template downloaded from the instructions for authors in Pharmaceuticals to ensure that our abstract complies with the journal's guidelines. So, it states: “Systematic reviews and original research articles should have a structured abstract of around 250 words and contain the following headings: Background/Objectives, Methods, Results, and Conclusions.” We also reviewed the abstract structure of articles already published in Pharmaceuticals and found that they use the same structure. Therefore, although we agree that an abstrac should not usually contain such headings, we must stick to the format requested by the journal.

Comment 2: Introduction. In my opinion, this section is too long. Also, the protective effect of phycobiliproteins from Arthrospira (Spirulina) maxima against cyclophosphamide-induced embryotoxicity and genotoxicity was studied for pregnant mice. Therefore, some general details about pregnancy in mice should be added in introduction, not only general things about pregnancy in humans.

Answer 2: Thank you for your feedback regarding the length of the introduction. While we understand that it may appear lengthy, our intention was to provide a comprehensive overview of the current situation surrounding cancer during pregnancy. This includes an exploration of the underlying mechanisms of action, therapeutic alternatives, their limitations, and the toxic effects associated with the most commonly used chemotherapeutic agents. We believe this detailed context is essential for a thorough understanding of our research objectives and the significance of our findings. By including this information, we aim to highlight the complexities and challenges faced in managing cancer during pregnancy, which ultimately justifies the need for our study. Therefore, we do not consider it appropriate to shorten the introduction, as it is critical for conveying the full scope of the issue at hand. Thank you for your understanding.

In other hand, we believe that your suggestions are very pertinent, so we have added the requested information on lines 51-58, which explain the unique aspects of pregnancy in CD1 mice versus humans. We believe this addition enhances the context of our research and provides a solid basis to understand our study. Thank you for your observation.

Comment 3: L414 -"In accordance with the literature..."at the end of the sentence should be added the reference where the four mechanisms are mentioned.

Answer 3: Thank you for your insightful comment. We share your belief in the importance of including references alongside relevant information, as it enhances the reader's understanding of the text. However, the paragraph in question is designed to provide a general overview (lines 438-441) before delving into a more detailed discussion of each of the four mechanisms mentioned (lines 441-453).

In this detailed description, we have included references that support each mechanism to facilitate quick access to the citations backing these points, minimizing the need for interpretation. While we appreciate your suggestion to add references to the initial statement regarding the four mechanisms, we believe that doing so would diminish their impact by lacking the necessary context and explanation in those initial lines.

Furthermore, including these references at the beginning would require us to reiterate them later in the text alongside their respective details, potentially leading to unnecessary redundancy. After careful consideration of your recommendation, we feel that maintaining our current structure is more appropriate for conveying the information effectively.

Comment 4: L498 - "Future studies should also investigate the long-term developmental outcomes in off- spring exposed to PBP and Cp during gestation, as well as the optimal dosing regimen for balancing protective effects with potential toxicity" In my opinion the limitations of the study should be clearly presented as limitations in the main text. I

Answer 4: We sincerely appreciate your observation. Although the limitations of our study and the suggestions for future research were already addressed in the final paragraph of the discussion, we agree with your valuable suggestion. So, we have completely rewritten this section to ensure that the limitations of our study are clearly and explicitly presented (lines 522–538). We believe this revised section now adequately emphasizes these important aspects. Thank you for helping us enhance the clarity and quality of our manuscript.

Comment 5: minor comments: L360 Table X?

Answer 5: Thank you for your observation. We acknowledge that there was an error in the text where “Table X” was written instead of “Table 4”. This mistake has now been corrected on line 384. We appreciate your careful review and attention to detail.

Reviewer 3 Report

Comments and Suggestions for Authors

The article entitled “Protective effects of phycobiliproteins from Arthrospira (Spirulina) maxima against cyclophosphamide-induced embryotoxicity and genotoxicity in pregnant CD1 mice” is recommended for publication after the following revisions:

1- The introduction is too long and should be written more concise

2- Given that the objectives of this study are to protect against CP in pregnancy, PBP did not provide complete protection based on the study results. Therefore, the overall objectives of the project are questionable. In the discussion of embryonic protection, the goal is 100% protection, which has not been achieved according to the results.

3- The LD50 of PBP should be mentioned in the discussion or introduction.

4- The study does not mention the amino acid sequence of PBP.

5- The reasons for investigating antioxidant activities should be clearly explained.

Comments on the Quality of English Language

no comments

Author Response

The article entitled “Protective effects of phycobiliproteins from Arthrospira (Spirulina) maxima against cyclophosphamide-induced embryotoxicity and genotoxicity in pregnant CD1 mice” is recommended for publication after the following revisions:

Comment 1: 1- The introduction is too long and should be written more concise.

Answer 1: Thank you for your insightful comment regarding the length of the introduction. While we appreciate your perspective, I would like to emphasize that the introduction was intentionally crafted to provide a comprehensive overview of the complexities surrounding cancer during pregnancy. This includes an exploration of the underlying mechanisms of action, limitations, relevant statistics, and the toxic effects associated with commonly used chemotherapeutic agents. Additionally, we offer a brief overview of potential alternatives, such as antioxidants, that may mitigate the side effects of chemotherapy, as well as a general description of phycobiliproteins.

Furthermore, it is important to note that another reviewer has requested additional information on a specific topic that is integral to the introduction. This feedback underscores the necessity of including detailed context to enhance the reader's understanding.

We believe that this level of detail is essential for grasping our research objectives and appreciating the significance of our findings. By presenting this information, we aim to illuminate the complexities and challenges involved in managing cancer during pregnancy, thereby justifying the need for our study. Consequently, we do not find it appropriate to shorten the introduction, as each section plays a critical role in conveying the full scope of this important issue. Thank you for your understanding.

Comment 2: 2- Given that the objectives of this study are to protect against CP in pregnancy, PBP did not provide complete protection based on the study results. Therefore, the overall objectives of the project are questionable. In the discussion of embryonic protection, the goal is 100% protection, which has not been achieved according to the results.

Answer 2: Thank you for your thoughtful observation regarding the objectives of our research. After careful consideration of your comments, we would like to clarify the following points:

In lines 197-200 of our manuscript, we state: “This study aimed to investigate the protective effect of phycobiliproteins (PBP), protein pigments with broad antioxidant capacity, isolated from Am, against cyclophosphamide (Cp)-induced damage during the embryonic development of CD1 mice, as well as their antigenotoxic effect in the mothers.” When we refer to evaluating the protective effect of PBP, we aim to assess their potential to prevent or reduce embryotoxic and genotoxic damage resulting from exposure to Cp during embryonic development. So, it is important to emphasize that we do not claim to demonstrate 100% protection against Cp-induced damage. As this is the first study of its kind, our primary objective is to establish that PBP can provide a degree of protection against Cp-related damage during pregnancy and to identify which tested doses are most effective. In essence, we aim to lay the groundwork for future research focused on this critical area.

Ultimately, our goal is to pave the way for new investigations that could lead to the development of complementary therapies for pregnant oncology patients, enhancing their quality of life and the health of their developing conceptus. In this context our results indicate a significant level of protection against embryotoxic and genotoxic damage induced by Cp, supporting our research objectives. Therefore, we do not believe that the objectives of our research are questionable in relation to what we were able to demonstrate. Thank you once again for your valuable feedback.

Comment 3: 3- The LD50 of PBP should be mentioned in the discussion or introduction.

Answer 3: Thank you for your valuable suggestion regarding the inclusion of the LD50 of PBP. We agree that this information is important for providing a comprehensive understanding of our study. As such, we have added this information in line 188 of the introduction. We appreciate your input and believe it enhances the clarity and depth of our manuscript.

Comment 4: 4- The study does not mention the amino acid sequence of PBP.

Answer 4: Thank you for your insightful comment regarding the amino acid sequence of PBP. We acknowledge that sequencing the PBP would have provided valuable information for our study. However, we did not perform this sequencing because the methodology we employed for the isolation and characterization of these proteins is well-established and has been in use for several decades. These methodologies are highly reproducible and continue to yield excellent results in current research. While we recognize that sequencing could have enhanced our investigation, budgetary constraints led us to prioritize other aspects of our study. Based on spectrophotometric analysis, we already had a solid understanding of the composition of our extract, which allowed us to focus on evaluating the protective effects of PBP rather than identifying their amino acid sequences. That said, we appreciate your suggestion and view it as a valuable direction for future research. We hope to explore this aspect in subsequent studies to further enrich our understanding of PBP. Thank you once again for your constructive feedback, which helps us improve our work.

Comment 5: 5- The reasons for investigating antioxidant activities should be clearly explained.

Answer 5:  Thank you for your insightful comment. In response to your feedback, we have revised the text to express this idea more clearly, ensuring a better understanding of the study's rationale and objectives (lines 189-197). We sincerely appreciate your suggestions, as they contribute to improving the quality and clarity of our manuscript.

Reviewer 4 Report

Comments and Suggestions for Authors

- The introduction is way too long and must be shortened. Also, subtitles should be removed  

- The extract should have either GC-MS or LC-MS done

- All comments are present in the attached PDF File. 

- The paper can only be accepted after doing the attached comments (major revision)

Comments on the Quality of English Language

Needs improvement. I have noticed some redundancy and grammatic errors. Please revise it regarding the language

Author Response

Comment 1: The introduction is way too long and must be shortened. Also, subtitles should be removed. The introduction needs to be reduced in half.

Answer 1: I sincerely appreciate your comment about the length of the introduction. I understand your concern and value your perspective. However, I would like to emphasize that given the nature of our research, the introduction was intentionally crafted to provide a comprehensive view of the complexities surrounding cancer during pregnancy. This includes an exploration of the underlying mechanisms of action, limitations, relevant statistics, and toxic effects associated with commonly used chemotherapeutic agents. In addition, we have included a brief summary of possible alternatives, such as antioxidants, that could mitigate the side effects of chemotherapy, as well as an overview of phycobiliproteins. We believe this information is crucial for the reader to understand the necessary context and relevance of our research. It is important to mention that another reviewer has also suggested adding additional information on a specific topic that we consider integral to the introduction. This feedback reinforces the need to include detailed context to enhance the reader's understanding. Therefore, we consider this level of detail is essential to capture our research objectives and appreciate the significance of our findings, as each section plays a critical role in conveying the magnitude of this topic. We appreciate your understanding and hope that this response meets your expectations and clearly conveys our intentions to keep our introduction to the same length.

Comment 2: Subtitles in the introduction should be removed

Answer 2: Thank you for your valuable comments regarding the use of subtitles in the introduction. We initially included these subtitles to enhance clarity by organizing the content into distinct sections. However, we understand your perspective and agree that removing them may improve the overall flow of the introduction. As a result, we have eliminated all subtitles from this section. Your insights are greatly appreciated, and we believe this change will contribute positively to the readability of our manuscript. Thank you once again!

Comment 3: The extract should have either GC-MS or LC-MS done

Answer 3: Thank you for your insightful comment regarding the use of GC-MS or LC-MS for our extract analysis. We fully recognize the immense value that these advanced techniques would provide in terms of separating, identifying, and quantifying the compounds present in our samples. However, we opted not to utilize these methods primarily due to the well-established spectroscopic techniques we employed for the isolation and characterization of PBP. These methodologies have been in use for several decades and continue to yield reliable and reproducible results, as supported by recent literature. The references we cited in our manuscript reflect contemporary studies that validate the efficacy of these techniques.

While GC-MS and LC-MS could potentially offer more precise data, we faced significant budgetary constraints that necessitated prioritization of other aspects of our research. So, our spectrophotometric analysis provided us with a solid understanding of the overall composition of our extract, allowing us to focus on assessing the protective effects of PBP without the additional complexity and cost associated with mass spectrometry.

Furthermore, it is important to note that the choice of analytical technique often depends on the specific objectives of the study and the nature of the samples being analyzed. In our case, the established methods were sufficient for achieving our research goals while maintaining accessibility and cost-effectiveness. We appreciate your suggestion and consider it a valuable direction for future investigations. We look forward to exploring more advanced analytical techniques in subsequent studies to further enrich our understanding of PBP extract. Thank you once again for your constructive feedback, which greatly aids in enhancing the quality of our work.

Comment 4: The whole paragraph starting with "Cancer is a disease, or a set of diseases ...... " and ending with "The most diagnosed cancers were lung cancer, ......... and stomach cancer'' should be removed.

Answer 4: We sincerely appreciate your valuable comments regarding the paragraph on cancer in our introduction. We consider it essential to maintain this section, as it provides an updated context that includes the most recent definition of the disease and relevant statistics highlighting its impact on global health. In particular, the data on cancer incidence and mortality offer the reader a more focused and relevant perspective that helps them understand the importance and scope of this disease. Although our work focuses on the most prevalent types of cancer among pregnant women. The data presented provide a solid basis for understanding the relevance and focus of our research. These elements underscore the seriousness of the problem and the urgent need to explore effective therapeutic alternatives.

So, we believe that deleting this paragraph could compromise clarity on the magnitude of the challenge we address and the relevance of our proposal. We thank you in advance for your understanding.

Comment 5: The authors' names are too many compared to the amount of work done. There should be a va lid explanation far this.

Answer 5:  Thank you for your comment regarding the number of authors on our manuscript. We understand that having multiple authors may raise questions about the contributions made to the work, and we appreciate the opportunity to clarify this matter. We want to assure you that the inclusion of several authors is not a reflection of any unethical practice. While our research may appear straightforward, it is rooted in a complex multidisciplinary framework that required expertise from various fields to effectively plan, develop, analyze, and interpret the results. Each author has played a significant role in their respective areas of expertise, contributing to different aspects of the study. Additionally, due to our limited resources, we sought assistance from researchers who provided essential guidance and access to equipment that were not initially included in our project scope. This collaboration was crucial for the successful execution of our research. Furthermore, we made a conscious decision to exclude some individuals who contributed less significantly to avoid inflating the author list unnecessarily. Every researcher listed as a co-author has made meaningful contributions to this study and deserves acknowledgment for their efforts. We appreciate your understanding and hope this explanation clarifies our rationale for the authorship structure in our manuscript. Thank you once again for your constructive feedback.

Comment 6:  Line 106, 107: "as a byproduct. Who is known" The two sentences need to be written in a correct way.

Answer 6: Thank you for your detailed review. We have considered your comment on lines 112 and 114 and corrected the sentence to improve its clarity and coherence. We believe this change improves the flow of the text and makes the idea easier to understand.

Comment 7: Figure 1 is very low in quality and resolution. lt needs to be replaced with a higher quality and resolution image.

Answer 7: Thank you for your observation. We appreciate your attention to detail, as high-quality visuals are essential for effectively conveying our research findings. In response to your comment, we have replaced this image with a higher quality and higher resolution version. We believe that this change ensures that the figure is now clearer and more visually appealing, allowing for better interpretation of the data presented. We are grateful for your constructive feedback, which has helped us to improve the overall quality of our manuscript. Thank you once again for your valuable input.

Comment 8: There should never be 3 paragraphs far statistical analysis. Please combine all the statistical analysis in one paragraph and under only 1 subtitle.

Answer 8: We greatly appreciate your comments and your interest in improving the clarity of our manuscript. We greatly value your comments and understand your point of view about consolidating the statistical analyses into a single paragraph. However, after careful consideration, we believe it is better to maintain the current structure of our statistical analysis sections; presenting the analyses individually, linked to each experimental section, facilitates the reader's understanding by establishing a direct connection between the experimental procedures and the results obtained. This linkage allows the reader to follow more intuitively the flow of the study and to understand how the conclusions were obtained. In addition, we believe that this structure facilitates the identification of the specific statistical methods used for each analysis, which is fundamental to the understanding of our study.

Comment 9: The extract should have either GC-Mass or LC-MS/MS.

Answer 9: Thank you for your insightful comment regarding the use of GC-MS or LC-MS for our extract analysis. We fully recognize the immense value that these advanced techniques would provide in terms of separating, identifying, and quantifying the compounds present in our samples. However, we opted not to utilize these methods primarily due to the well-established spectroscopic techniques we employed for the isolation and characterization of PBP. These methodologies have been in use for several decades and continue to yield reliable and reproducible results, as supported by recent literature. The references we cited in our manuscript reflect contemporary studies that validate the efficacy of these techniques.

While GC-MS and LC-MS could potentially offer more precise data, we faced significant budgetary constraints that necessitated prioritization of other aspects of our research. So, our spectrophotometric analysis provided us with a solid understanding of the overall composition of our extract, allowing us to focus on assessing the protective effects of PBP without the additional complexity and cost associated with mass spectrometry.

Furthermore, it is important to note that the choice of analytical technique often depends on the specific objectives of the study and the nature of the samples being analyzed. In our case, the established methods were sufficient for achieving our research goals while maintaining accessibility and cost-effectiveness. We appreciate your suggestion and consider it a valuable direction for future investigations. We look forward to exploring more advanced analytical techniques in subsequent studies to further enrich our understanding of PBP extract. Thank you once again for your constructive feedback, which greatly aids in enhancing the quality of our work.

Comment 10: References 55 & 61 are too old. Please replace with more recent references.

Answer 10: We sincerely appreciate your interest in improving the quality and timeliness of our manuscript. We would like to clarify that the references mentioned (55 and 61) correspond to the original research that developed the methodology used in our study. We consider it important to acknowledge the authors whose contributions continue to be pillars in this field. These calculations although proposed long ago, are still widely accepted and used in recent research.  In the case of reference 55 (now 57) in addition to the original citation, we had already included recent references that apply this formula ([54,55] line 562), which reinforces its current relevance. Regarding reference 61 (now 63), we have followed your valuable recommendation and added an additional more current reference ([64] line 636). This adjustment strengthens the timeliness of the sources without losing sight of the importance of the original work. In addition, it contributes to the understanding of the modifications and/or variations that have occurred in the formulas over time. Although in essence they are still the same formulas, changes have been made according to the needs of researchers that could confuse the reader. For this reason, we allow ourselves to justify the permanence of the original citations 57 and 63, in addition to recent references. We thank you for your comments, which allowed us to further enrich and update our manuscript. We trust that this revision will adequately address your comments and reinforce the soundness of our work.

Round 2

Reviewer 3 Report

Comments and Suggestions for Authors

no comments

Author Response

Thank you for taking the time to review our manuscript. We carefully analyzed your review report, but we did not find any specific comments or suggestions to address. Therefore, we assume that you are in agreement with the final version of the text.We appreciate your support and look forward to your continued feedback.

Reviewer 4 Report

Comments and Suggestions for Authors

The paper can be accepted in its current form

Author Response

Thank you very much for your positive feedback and for recommending the acceptance of our paper in its current form. We truly appreciate your time and effort in reviewing our work, as well as your insightful comments that have contributed to enhancing the quality of the manuscript. We are grateful for your support and look forward to the opportunity to share our findings with the broader community.